# EUS-FNA versus EUS-FNB in Pancreatic Solid Lesions ≤ 15 mm

**DOI:** 10.3390/diagnostics14040427

**Published:** 2024-02-15

**Authors:** Maria Cristina Conti Bellocchi, Micol Bernuzzi, Alessandro Brillo, Laura Bernardoni, Antonio Amodio, Nicolò De Pretis, Luca Frulloni, Armando Gabbrielli, Stefano Francesco Crinò

**Affiliations:** 1Diagnostic and Interventional Endoscopy of the Pancreas, Pancreas Institute, G.B. Rossi University Hospital, 37134 Verona, Italy; laura.bernardoni@aovr.veneto.it (L.B.); stefanofrancescocrino@aovr.veneto.it (S.F.C.); 2Department of Medicine, Pancreas Institute, G.B. Rossi University Hospital, 37134 Verona, Italyalessandro.brillo@studenti.univr.it (A.B.); antonio.amodio@aovr.veneto.it (A.A.); nic_depretis@yahoo.it (N.D.P.); luca.frulloni@univr.it (L.F.)

**Keywords:** endoscopic ultrasound, EUS-guided fine needle aspiration, EUS-guided fine needlebiopsy, pancreatic cancer, neuroendocrine tumors, solid pancreatic lesions

## Abstract

A small tumor size may impact the diagnostic performance of endoscopic ultrasound-guided tissue acquisition (EUS-TA) for diagnosing solid pancreatic lesions (SPLs). We aimed to compare the diagnostic yield of EUS-guided fine-needle aspiration (FNA) and biopsy (FNB) in SPLs with a diameter ≤ 15 mm. Consecutive patients who underwent EUS-TA for SPLs ≤ 15 mm between January 2015 and December 2022 in a tertiary referral center were retrospectively evaluated. The primary endpoint was diagnostic accuracy. The final diagnosis was based on surgical pathology or disease evolution after a minimum follow-up of 6 months. Inadequate samples were all considered false negatives for the study. Secondary outcomes included sample adequacy, factors impacting accuracy, and safety. We included 368 patients (52.4% male; mean age: 60.2 years) who underwent FNA in 72 cases and FNB in 296. The mean size of SPLs was 11.9 ± 2.6 mm. More than three passes were performed in 5.7% and 61.5% of patients in the FNB and FNA groups, respectively (*p* < 0.0001). FNB outperformed FNA in terms of diagnostic accuracy (89.8% vs. 79.1%, *p* = 0.013) and sample adequacy (95.9% vs. 86.1%, *p* < 0.001). On multivariate analysis, using FNA (OR: 2.10, 95% CI: 1.07–4.48) and a final diagnosis (OR: 3.56, 95% CI: 1.82–6.94) of benign conditions negatively impacted accuracy. Overall, the adverse event rate was 0.8%, including one pancreatitis in the FNA group and one pancreatitis and one bleeding in the FNB group, all mild and conservatively managed. EUS-TA for SPLs ≤ 15 mm has a high diagnostic yield and safety. This study suggests the superiority of FNB over FNA, with better performance even with fewer passes performed.

## 1. Introduction

Endoscopic ultrasound with tissue acquisition (EUS-TA) using fine-needle aspiration (FNA) or, more recently, fine-needle biopsy (FNB) needles has become an indispensable tool for the diagnosis of solid pancreatic lesions (SPLs) with a sensitivity of 87–94% and a specificity close to 100% [1]. Several factors have already been shown to influence the diagnostic accuracy of EUS-TA, such as the use of rapid on-site evaluation (ROSE) for immediate cytopathologic assessment [2]; needle features (type [3] and caliber [4]); procedure-related aspects (number of passes [5], sampling technique [6], and use of suction [7,8]); and lesion characteristics, including size [9]. Due to the widespread use of cross-sectional abdominal imaging and recent advances in imaging modalities, small SPLs are increasingly being found, making accurate pathologic diagnosis essential for proper patient management. For pancreatic ductal adenocarcinoma (PDAC), the tumor stage at the time of diagnosis is the most critical factor affecting disease prognosis. Although detection at an early stage is rare, it has been associated with better survival [10]. Conversely, a definitive diagnosis of a benign disease (e.g., chronic pancreatitis or autoimmune pancreatitis) guarantees appropriate management and avoids unnecessary surgery.

Few studies have addressed the accuracy of EUS-TA for the histopathologic diagnosis of small SPLs [11,12,13], reporting contrasting results. Moreover, the effect of lesion size on diagnostic accuracy has been obtained from a large series after stratifying patients by lesion diameter [13]. In a meta-analysis of 33 studies with 6883 cases [9], the diagnostic yield of EUS-TA was lower for SPLs ≤ 20 mm than that of SPLs > 20 mm in terms of sensitivity [odds ratio (OR): 3.05, 95% confidence interval (CI): 1.25–7.42; *p* = 0.01] and accuracy (OR: 3.27, 95% CI: 1.55–6.89; *p* < 0.01). In addition, this meta-analysis showed that none of the factors, including the use of FNB needles, the slow pull technique, or ROSE, improved the accuracy of EUS-TA for small SPLs. Only the use of a 25-gauge needle slightly improved the sensitivity of EUS-TA for small SPLs but without reaching statistical significance. However, most of the included studies refer to FNA needles, and data on the performance of FNB in small SPLs are lacking. Moreover, while PDAC represents the most common diagnosis in EUS-TA studies, small SPLs are often of different etiologies [14], with a higher prevalence of pancreatic neuroendocrine tumors (Pan-NETs) and other lesions requiring specific management based on preoperative diagnosis [15]. Finally, some authors have suggested a higher risk of adverse events (AEs) when small SPLs are punctured [16].

In this study, we aimed to evaluate the diagnostic yield of both FNA and FNB in defining small SPLs with a diameter ≤ 15 mm.

## 2. Materials and Methods

### 2.1. Patients

Consecutive adult patients referred to our endoscopy department with SPL ≤ 15 mm between January 2015 and December 2022 were retrospectively evaluated. Patients with larger tumors, cystic lesions, and the inability to perform EUS-TA due to a not-visible target or contraindication to puncture or technical failure, as well as patients included in randomized trials with a specific sampling protocol, were excluded. Patients without at least six months of follow-up dropped out. Written informed consent was obtained from all patients for the procedure following the Declaration of Helsinki. This study was approved by our institution’s ethics committee.

### 2.2. Aims

The study’s primary objective was to compare the diagnostic accuracy of EUS-FNA and EUS-FNB for diagnosing SPLs ≤ 15 mm. Other diagnostic measures (i.e., sensitivity, specificity, negative predictive value (NPV), and positive predictive value (PPV)) were also evaluated. Moreover, a subgroup analysis including lesions ≤ 10 mm was performed. 

Secondary objectives included adequacy rate and safety, as well as factors affecting accuracy, including patient demographics (gender and age), lesion characteristics [size, location, final diagnosis, and use of contrast-harmonic EUS (CH-EUS)] [17], and procedure-related features (type and caliber of needle and number of passes). Moreover, the feasibility of the Ki-67 index in the subgroup of patients with Pan-NETs was evaluated [18].

### 2.3. Procedures

EUS examinations were performed by experienced endosonographers using a linear array echoendoscope (EG-3870UTK, Pentax Medical, Tokyo, Japan) under deep sedation with anesthesiologic assistance. Over the study period, three types of needles with a 22- or 25-gauge caliber have been used: (1) standard FNA needle (Expect™, Boston Scientific Corp, Boston, MA, USA); (2) side-fenestrated needle (Echo Tip ProCore™ HD, Cook Medical, Limerick, Ireland); and (3) end-cutting needles (SharkCore™, Medtronic, Minneapolis, MI, USA; Acquire™, Boston Scientific Corp, Boston, MA, USA). The needle type was chosen according to availability, and the size (22- or 25-gauge) was chosen at the discretion of the endosonographer. The 25-gauge caliber was preferred when the endoscope was in a torqued or unstable position, when >10 mm of normal pancreas needed to be traversed, or when the distance between the lesion and the transducer was > 15 mm. For the purpose of the study, the two end-cutting needles were grouped based on the results of a recent meta-analysis demonstrating comparable performance between SharkCore™ and Acquire™ [1]. Moreover, lesions sampled with a 25-gauge side-fenestrated needle were included in the FNA group, whereas those sampled with a 22-gauge side-fenestrated needle were included in the FNB group. The fanning sampling technique and the slow-pull technique were routinely performed. The number of passes was left to the endoscopist’s decision, influenced by several factors, including the stability of the scope, the deep location of the lesion, the possible occurrence of AEs related to the interposition of the main pancreatic duct or vessels, and the use of macroscopic on-site evaluation (MOSE) [19]. The collected samples using an FNA or a 25-gauge side-fenestrated needle were processed as cell blocks [20]. Samples collected using an FNB needle were placed directly in a formalin vial and handled as standard histological specimens. ROSE was not performed in any of the cases. The histological classification of EUS samples was based on the Papanicolaou classification [21].

### 2.4. Definitions

Diagnostic accuracy was defined as the percentage of lesions sampled that matched the final diagnosis in terms of “malignancy” (i.e., correct classification as benign or malignant). Low-grade tumors (e.g., Pan-NET or solid pseudopapillary neoplasm) were considered malignant for the study. The primary endpoint was assessed using “strict” criteria (i.e., specimens reported as “atypical” and “suspicious for malignancy” were considered negative, and only specimens reported as “malignant” were categorized as positive). Additional analyses were performed using “non-strict” criteria, thus classifying “suspicious for malignancy” samples as positive for malignancy.

The final diagnosis was assessed on surgical specimens whenever available. In non-resected patients, it was established based on the diagnostic workup (combined results of imaging studies and any additional biopsy results) and the clinical course of the disease for at least six months [22]. Specimens with insufficient material were considered false negatives. 

Adequacy was defined as the percentage of lesions sampled in which the obtained material is representative of the target site and sufficient for diagnosis [22]. 

Adverse events were defined according to the American Society of Gastrointestinal Endoscopy lexicon [23].

### 2.5. Statistical Analysis

Patients’ characteristics were summarized by descriptive statistics [mean ± standard deviation (SD) or median with interquartile range (IQR) for continuous variables and frequency distributions for categorical variables]. The overall diagnostic accuracy and adequacy rate of FNA and FNB were calculated and compared. The chi-square test and the unpaired *t*-test were used to compare categorical and continuous data, respectively. Factors affecting the diagnostic accuracy of EUS-TA were analyzed using univariate and multivariate analyses. Multivariate analysis was carried out employing logistic regression with a backward stepwise technique. The coefficients obtained from the logistic regression analysis were expressed in terms of odds ratios (ORs). We included variables that impact diagnostic accuracy (i.e., lesion size and location, use of CH-EUS, needle type and caliber, number of passes, and final diagnosis). Covariates with a *p*-value < 0.10 in the univariate analysis were subsequently included in the multivariate analysis. All analyses were performed using SPSS, with a statistical significance level of 5% and corresponding 95% CIs. A two-sided distribution was used, and statistical significance was assumed at *p* < 0.05.

## 3. Results

Between January 2015 and December 2022, 426 patients met inclusion/exclusion criteria. A total of 51 patients were excluded: 40 because EUS-TA was not performed, and 11 because they were included in randomized controlled trials. Moreover, 7 patients dropped out because of a lack of follow-up. The reasons for not performing the puncture were proximity to the main pancreatic duct (24.5%), interposed vessels on the needle route (24.4%), lack of indication for spontaneous size reduction (19.5%), deep location (17%), technical failure (7.3%), or abortion of the EUS examination because of intraprocedural complications (7.3%, including two sedation-related AEs and one duodenal perforation). Therefore, 368 patients (52.4% male; mean age: 60.2 ± 13.1 years) were included in the study, of whom 72 were punctured with FNA and 296 with FNB. The flow diagram is shown in Figure 1.

### 3.1. Pre-Procedural Data

Baseline data are shown in Table 1.

Patients were mainly asymptomatic, and small SPLs were found incidentally. Imaging procedures performed before EUS included transabdominal ultrasound (39.7%), computed tomography scan (CT) (82.3%), and magnetic resonance imaging (MRI) (67.9%). In addition, 18F-FDG-PET was performed in 31.8% of cases and 68Ga-DOTATOC-PET in 42.4% [24]. The mean size of the lesion based on cross-sectional imaging was 13.4 ± 5.1 mm. Suspected PDAC or small metastatic lesions on cross-sectional imaging were reported in 73 (19.8%) patients and confirmed in 56 (76.7%) cases. In comparison, the most frequently reported possible diagnosis was Pan-NET in 182 cases (49.5%), which was eventually confirmed in 153 (84.1%). Serous Ca 19.9 level was elevated only in 10% of patients with an available test.

### 3.2. EUS Findings

On EUS examination, the average lesion size was 11.9 ± 2.6 mm, slightly smaller than that reported on radiological imaging. In 10 patients without evidence of a lesion on MRI or CT but showing indirect signs (dilation of the main pancreatic duct or positive PET), a nodule was found on EUS examination (Figure 2), with a mean diameter of 11.8 mm (range 5–15 mm).

PDAC, Pan-NET, and chronic pancreatitis were diagnosed in four (40.0%), four (40.0%), and two (20.0%) of them, respectively. CH-EUS evaluation was performed in 90.0% of patients, with a hypervascular pattern observed in 180 (48.9%) cases and a hypovascular appearance in 105 (28.5%). The EUS findings are shown in Table 1.

### 3.3. Final Diagnoses

The final diagnosis was assessed on surgical specimens in 93 out of 368 patients (25.3%) and based on the evolution of the disease in 275 cases (74.7%). In non-resected patients, the mean follow-up period was 30.3 ± 21.8 months (range 6–97), and a malignant disease was diagnosed in 221 cases (80.0%; Pan-NET, PDAC, and metastases were diagnosed in 174, 24, and 22 patients, respectively, while one patient had a diagnosis of lymphoma), whereas a benign/inflammatory condition was diagnosed in 49 cases (17.8%); in the remaining five patients (1.8%), the diagnosis was undetermined because they were lost to follow-up after an unconclusive result of EUS-TA (Table 2).

### 3.4. Diagnostic Accuracy

Overall, the diagnostic accuracy of EUS-TA in small SPLs was 88.0% (95% CI: 84.2–91.1). When considering 93 patients who underwent resection, diagnostic accuracy was 88.2%, similar to that of non-resected patients (89.1%). Twelve EUS samples were classified as “suspicious” for malignancy and were eventually proven to be from malignant diseases. When “strict criteria” were applied, 16 and 30 cases were categorized as false negatives in the FNA and FNB groups, respectively, resulting in diagnostic accuracy of 79.1% (95% CI: 67.9–87.8) for FNA and 89.8% (95% CI: 85.5–93.1) for FNB (*p* = 0.013). Similar results were observed when “non-strict” criteria were used (i.e., when “suspicious samples” were considered positive for malignancy). Overall accuracy increased to 90.7% (95% CI: 87.3–93.5) and was significantly higher in the FNB group [93.2% (95% CI: 89.7–95.8) vs. 80.5% (95% CI: 69.5–88.9) for FNA (*p* = 0.0009)]. Sensitivity, specificity, NPV, and PPV using strict and non-strict criteria are reported in Table 3 and Table 4.

We further performed a subgroup analysis of patients with lesions ≤ 10 mm (n = 114), and the superiority of FNB (n = 88) over FNA (n = 26) was confirmed in terms of accuracy (95.4% vs. 77.8%; *p* = 0.011). Moreover, we evaluated the sensitivity of EUS-FNA and EUS-FNB in the subgroup of patients with PDAC. We found a slightly lower sensitivity after EUS-FNA compared with EUS-FNB, although it was not statistically significant (60.0% vs. 79.4%, *p* = 0.228).

### 3.5. Diagnostic Adequacy

Overall, EUS-TA provided an adequate specimen for cytohistologic evaluation in 346 out of 368 (94.0%) patients. Diagnostic adequacy was significantly higher in the FNB group than in the FNA group (95.9% vs. 86.1%, *p* < 0.001). Figure 3 reports an example of histopathological evaluation of the material collected using FNA and FNB. Among 22 inadequate samples (12 FNB and 10 FNA), five (22.7%) patients were lost on follow-up, three (13.6%) underwent surgery [with a final diagnosis of PDAC (n = 1), Pan-NET (n = 1), and metastasis of renal cancer (n = 1)], two (9.1%) had a radiological diagnosis of Pan-NET, whereas two (9.1%) had a diagnosis of PDAC and two (9.1%) a diagnosis of metastases based on further biopsy samples. No disease progression was documented during follow-up in the remaining eight (36.4%) patients, and a benign/inflammatory condition diagnosis was assessed.

### 3.6. Safety

Overall, only three (0.8%) mild AEs were described. One retroperitoneal hematoma occurred during the EUS-FNB of a small metastasis of renal cancer, requiring prolongation of hospital stay for clinical monitoring and being conservatively managed with no need for reintervention. Two mild pancreatitis were recorded after puncturing two small SPLs: one Pan-NET in the FNA group and one PDAC close to the main pancreatic duct in the FNB group. Both patients were treated conservatively and discharged within three days.

### 3.7. Factors Impacting Diagnostic Accuracy

To clarify factors affecting the diagnostic accuracy of EUS-TA in small SPLs, uni- and multivariate analyses were conducted (Table 5).

On univariate analysis, FNA, benign final diagnoses, and pancreatic head location negatively impacted accuracy. Although a trend toward significance was observed for lesion sites, only FNA and benign final diagnoses were independent factors in multivariate analysis. No differences were found for hypo- or hypervascular lesions, nor for the different needle calibers.

### 3.8. Feasibility of Ki-67 in Pan-NETs

Among patients with an adequate specimen, a diagnosis of Pan-NET was established in 196 (53.2%) patients: 40 punctured with FNA and 156 with FNB. Overall, Ki-67 index evaluation was feasible in 185 out of 196 lesions (94.3%) without differences between the two groups (90.0% in the FNA group vs. 95.5% in the FNB group, *p* = 0.239). When performed, according to the WHO classification, the Ki-67 value was ≤3% in 175 (94.6%) and >3% in 10 (5.4%) cases, with four cases showing a Ki-67 value higher than 5%. A total of 30 out of 206 (15.5%) patients with Pan-NETs underwent resection. Tumor grading based on the Ki-67 value assessed on EUS specimens was consistent with surgical samples in 26 (86.7%) cases. In the remaining 4 (13.3%) cases, an undergrading (G1 on EUS samples vs. G2 on surgical specimens) was observed.

## 4. Discussion

The detection of small SPLs is increasing due to the widespread availability of high-quality cross-sectional imaging. EUS-TA is considered the diagnostic procedure of choice, with FNA and FNB equally recommended for tissue sampling [25]. However, this recommendation comes from studies mainly including large-size tumors without subgroup analyses in relation to the size of the lesions. Moreover, the diagnostic yield of EUS-TA for small SPLs seems disappointing [9], with several studies showing that the size of SPLs may influence its performance. In a large series of SPLs, the lesion size was an independent factor affecting the accuracy of EUS-FNA, together with the head location of the lesion, the availability of ROSE, and the final diagnosis (benign vs. malignant) [26]. Siddiqui et al. [13], in a cohort of 583 patients with SPLs of various sizes, showed that diagnostic accuracy and sensitivity are strongly correlated with tumor size, namely, 40% and 47%, respectively, for lesions ≤10 mm, compared to 91.6% and 88% observed in the group of lesions > 40 mm. Unfortunately, the number of cases in each group was not reported. In a previously published series from our center where EUS-FNA of SPLs was performed with a 25-gauge standard needle, the lesion size was the only independent factor related to diagnostic accuracy, ranging from 80.6% to 92.5% when the diameter was less than 15 mm and more than 25 mm, respectively [11]. However, whether new-generation FNB needles may improve diagnostic accuracy for small SPLs is unknown. 

To answer this question, we performed a large retrospective study including more than 360 patients with small SPLs and compared the outcomes of FNA with those of FNB. We demonstrated that FNB outperformed FNA in terms of diagnostic accuracy (89.8% for FNB vs. 79.1% for FNA, *p* = 0.013) and sample adequacy (95.9% for FNB vs. 86.1% for FNA, *p* < 0.001). Two points concerning our study should be underlined. First, in the FNB group, we used a new-generation end-cutting needle in approximately 95% of cases. Second, in the FNA group, ROSE was never available. The performance of end-cutting needles in small SPLs is poorly known because previous studies mainly employed a side-fenestrated needle. In particular, Fabbri et al. [27] evaluated the role of 22-gauge side-fenestrated needles in 68 patients with SPLs < 20 mm, and 82% accuracy was found. Therefore, the 90% accuracy we found in our study seems significantly better, even if we used a smaller-caliber needle (25-gauge) in 65% of cases. In the paper by Mie et al. [28], Franseen needles were compared to 22-gauge FNA needles and 20-gauge forward-bevel needles in small (<20 mm) SPLs. The accuracy of the Franseen needle was 86%, significantly lower when compared to 93% obtained with FNA needles and 97% obtained with the forward-bevel FNB needle, postulating the unsuitability of the “three-cutting surfaces” tip needle for the evaluation of small SPLs. This result contrasts with the present study, where an excellent diagnostic accuracy of FNB was found. This difference may be explained by the use of a different end-cutting needle in most of the patients in the present study. Indeed, we mainly used the SharkCore™ needle, which differs from the Franseen one because of the asymmetric shape of the tip, which is characterized by a longer penetrating tip, ensuring excellent penetration capability. In small tumors, needle penetration is crucial to obtaining a satisfactory sample because the distance range of the fanning movements is limited. In a recently published study about the performance of repeated EUS-FNB after previous nondiagnostic or inconclusive sampling [29], both a lesion size > 23 mm (OR: 3.04, 95% CI: 1.31–7.06; *p* = 0.009) and the use of a second-generation end-cutting FNB needle (OR: 5.42, 95% CI: 2.30–12.77; *p* < 0.001) were independently related to sample adequacy in repeated EUS-FNB. In this study, the diagnostic accuracy was affected by lesion size (OR: 1.03, 95% CI: 1.00–1.06; *p* = 0.05) and repeated sampling at high-volume centers (OR: 2.12, 95% CI: 1.10–3.17; *p* = 0.03) on multivariate analysis. However, in a recent network meta-analysis, the Franseen and SharCore™ needles had comparable performance for tissue sampling of pancreatic masses, mainly when a 22-gauge caliber needle was used, regardless of the lesion size [1]. Therefore, further studies comparing the SharkCore™ and Franseen needles in the setting of small SPLs are needed to draw definitive conclusions.

On the other hand, some authors suggest that SPL size may not affect the performance of FNA, particularly in the presence of ROSE. Ramesh et al. retrospectively compared the performance of EUS-FNA with ROSE in 581 SPL patients, with promising results [30]. However, about 70% had a lesion with a diameter > 20 mm, only 15 patients had lesions with a diameter < 10 mm, and 4 were false negatives [30]. Despite the potential advantage of ROSE in specific settings, including small SPLs or previously nondiagnostic EUS-TA results, its role during EUS-TA remains controversial. It might not be associated with a real improvement in diagnostic yield, as described in two meta-analyses [31,32]. In addition, the role of ROSE seems to have decreased in the era of FNB needles. Since their introduction, FNB needles, and particularly end-cutting needles, have shown their superiority in comparison to FNA, making ROSE less useful, as shown in a large randomized controlled trial that demonstrated that FNB performed without ROSE is not inferior to FNB performed with ROSE [2]. However, further studies are needed to assess if ROSE should be employed to diagnose small SPLs, regardless of whether FNA or FNB needles are used.

In our study, both 22- and 25-gauge needle calibers were used. Unlike the meta-analysis of Nakai et al. [9], where a potential advantage of 25-gauge needles in small SPLs had been observed (despite not being statistically significant), we did not observe any impact of needle caliber on diagnostic accuracy. Our finding agrees with what was reported in another meta-analysis on this topic [33]. Moreover, a prospective randomized study comparing the performance of 25- and 22-gauge Franssen needles demonstrated comparable histologic adequacy and diagnostic accuracy using these two different needle calibers [34]. In our opinion, the choice of needle caliber should be left to the endoscopist’s preference and based on the puncture route or difficult/deep lesion location rather than the histologic yield expected. Finally, we observed the superiority of FNB over FNA despite a lower number of passes performed in the FNB group. This represents a great advantage of new-generation needles that guarantee better tissue core procurement with a mean number of passes of two in small SPLs and without ROSE, as recently reported [9,35].

On multivariate analysis, in addition to using FNA, a final diagnosis of a benign condition and head/uncinate site have also been shown to impact accuracy negatively. The location of the lesion in the head/uncinate process has been reported as negatively affecting the accuracy of EUS-FNA in a previous study [26]. Probably, the body–tail location is a more accessible site for a puncture. In contrast, the head–uncinate process is more challenging due to the scope position or the lesion’s deep location, especially in small SPLs. The malignancy rate in our cohort was about 85% when low-grade neoplasms (e.g., Pan-NETs) were also considered malignant. Nevertheless, the adequacy rate was lower in patients with a benign condition, probably because of a worse definition of margins in benign lesions (e.g., chronic pancreatitis), thus resulting in a more difficult target identification, especially in small lesions. 

In small SPLs, Pan-NET incidence is not neglectable [15]. Precise diagnosis and risk stratification are crucial for proper management, with preoperative cyto/histology being one of the most important findings for this purpose [15,36]. The EUS-TA result may significantly impact the management of small (≤20 mm) Pan-NETs. Guidelines recommend surgical resection in cases of grade 2 Pan-NETs, even with a diameter ≤ 20 mm [37]. In the present study, despite a not statistically significant difference observed between FNA and FNB regarding Ki-67 assessment feasibility, in the FNB group, we reached approximately 96% of suitable samples compared with 90% in the FNA group. This result agrees with previous studies [38]. Probably, the feasibility of immunostaining could result in it being easier on histological specimens than on cytological ones [39].

Moreover, the present study observed a concordance rate of approximately 87% between EUS-FNA/B and surgical specimens, which agrees with a recent meta-analysis [18]. Finally, new immunohistochemical markers that predict Pan-NETs prognosis are feasible and reliable on FNB specimens [40]. In the future, large studies should evaluate the impact of such markers on the risk stratification of small Pan-NETs and the possibility of including less invasive treatments to manage these lesions [41,42].

In a Japanese study by Katanuma et al. [16], tumor size ≤ 20 mm (OR: 18.48, 95% CI: 3.55–96.17) and the diagnosis of Pan-NETs (OR: 36.50, 95% CI: 1.73–771.83) were independent risk factors for procedural complications of EUS-FNA (with an overall rate of 14.2%). However, four out of eight AEs described were mild abdominal pain that, according to the American Gastrointestinal Endoscopy Association lexicon, could be considered “incidents” if additional intervention or prolonged hospitalization is not required [23]. In contrast, in our study, where AEs were defined using standardized definitions [23], the rate of AEs was lower than 1%, and EUS-TA was demonstrated to be a safe procedure even in small-size lesions. 

To our knowledge, our study is the largest series of EUS-guided cyto-histopathological diagnoses of small SPLs and the first to compare FNA and FNB in this setting. The FNB diagnostic accuracy was close to 90% and seems comparable to that reported in studies including larger-diameter lesions, thus suggesting that FNB should be preferred over FNA in this subgroup of patients. However, our study has several limitations. First, the retrospective design and the absence of randomization carry selection biases that can be difficult to mitigate. Moreover, a larger number of cases were included in the FNB group compared with the FNA group. Second, we observed a 10% dropout rate because of non-punctured lesions that could have impacted the results. Third, this study was performed in a single tertiary center with high expertise in biliopancreatic EUS and may not be reproducible in different settings. Fourth, we used different needle types in the FNB groups, and we included the patients sampled with a 25-gauge ProCore™ needle in the FNA group. Therefore, differences related to the tip design cannot be demonstrated. Moreover, in our study, more than 22G-caliber needles were used in the FNB group compared with the FNA group, where almost all lesions were sampled with a 25G needle. However, needle caliber was not associated with diagnostic accuracy in multivariate analysis, which agrees with a recent meta-analysis that demonstrated no differences between 22G and 25G needles [33]. Fifth, a cut-off of 20 mm is commonly used for pancreatic tumor staging, slightly different from the 15 mm one we used in the present study. Finally, we did not use ROSE, and its role could not be evaluated.

In conclusion, EUS-TA for small (≤15 mm) SPLs has a high diagnostic yield and has been proven safe. This study showed the superiority of FNB over FNA, with better performance in terms of both adequacy and diagnostic accuracy, even in benign lesions, regardless of the caliber of the needle used and with a lower number of passes.

## Figures and Tables

**Figure 1 diagnostics-14-00427-f001:**
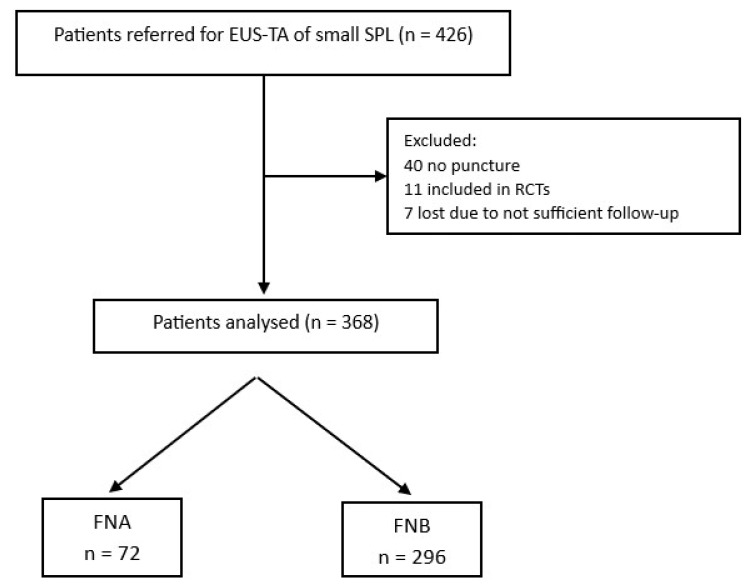
Study flowchart. EUS-TA: Endoscopic UltraSound Tissue Acquisition; SPL: solid pancreatic lesions; RCTs: randomized controlled trials; FNA: fine needle aspiration; FNB: fine needle biopsy.

**Figure 2 diagnostics-14-00427-f002:**
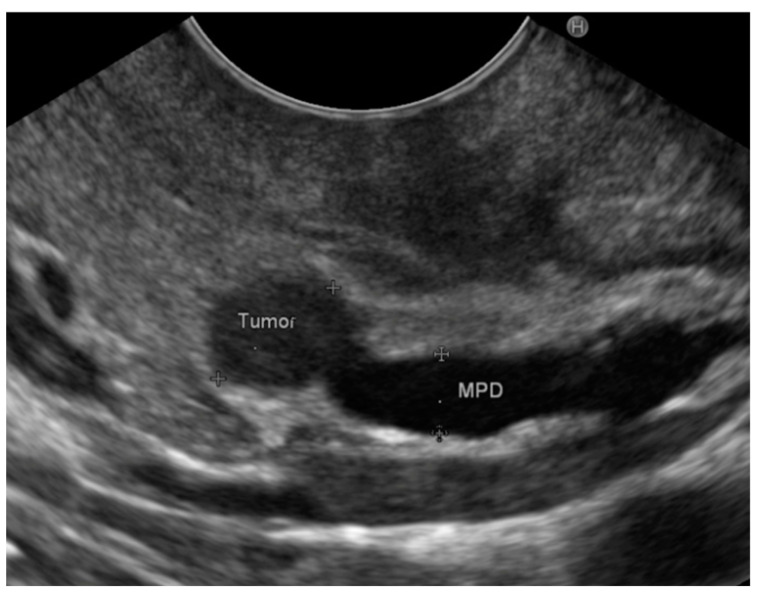
Small Pan-NET of the pancreas with upstream dilation of the main pancreatic duct.

**Figure 3 diagnostics-14-00427-f003:**
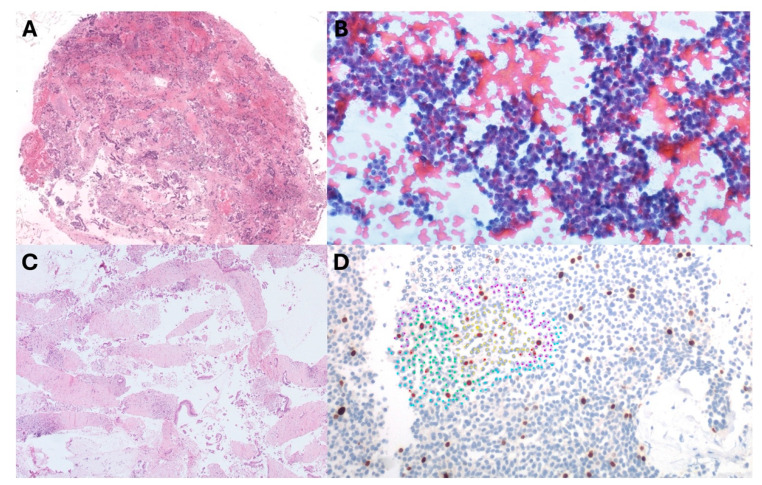
Examples of specimens collected with fine-needle aspiration (**A**,**B**) and fine-needle biopsy (**C**,**D**). (**A**) In a dirty, necrotic background, evaluable aggregates of adenocarcinoma cells can be recognized. (**B**) A fine-needle aspiration from a pancreatic neuroendocrine tumor shows small cells with eccentric nuclei and coarse chromatin. (**C**) Strands of cells and tissue fragments of adenocarcinoma are intermingled with desmoplastic stroma. (**D**) Fragments of a neuroendocrine tumor, solid pattern. Ki-67 immunolabelling identifies the replicative cells that show brown nuclei.

**Table 1 diagnostics-14-00427-t001:** Baseline characteristics and endoscopic ultrasound details of 368 patients who underwent endoscopic ultrasound-guided tissue acquisition of a small pancreatic solid lesion.

	Overall n = 368	FNA n = 72	FNB n = 296	*p*-Value
Age (years), mean ± SD	60.2 ± 13.1	58.8 ± 13.3	60.5 ± 13.1	0.335
Sex, N (%)				0.264
Male	193 (52.4%)	42 (58.3%)	151 (51.0%)	
Female	175 (47.6%)	30 (41.7%)	145 (49.0%)	
Clinical onset, N (%)				0.845
Incidental finding	151 (41.0%)	31 (43.1%)	120 (40.5%)	
Oncological follow-up	63 (17.2%)	12 (16.7%)	51 (17.2%)	
Abdominal pain	63 (17.2%)	15 (20.8%)	48 (16.2%)	
Hypoglycemia	30 (8.1%)	3 (4.2%)	27 (9.1%)	
Dyspepsia	24 (6.5%)	4 (5.6%)	20 (6.7%)	
Jaundice	21 (5.7%)	4 (5.6%)	17 (5.7%)	
Weight loss	8 (2.2%)	3 (4.2%)	5 (1.7%)	
Acute pancreatitis	6 (1.6%)	0 (0%)	6 (2.0%)	
Hyperenzymemia	2 (0.5%)	0 (0%)	2 (0.7%)	
Ca19.9, N (%)				0.542
Not available	220 (59.8%)	55 (76.4%)	165 (55.7%)	
Positive *	34 (9.3%)	5 (7.0%)	29 (9.8%)	
Negative	114 (31.0%)	12 (16.6%)	102 (34.5%)	
Size on cross-sectional imaging (mm),				0.598
mean ± SD **	13.4 ± 5.1	11.8 ± 2.4	13.3 ± 5.0	
Size on EUS (mm),				
≤5	5 (1.4%)	0 (0%)	5 (1.7%)	
6–10	109 (29.6%)	26 (36.1%)	83 (28.0%)	0.356
11–15	254 (69.0%)	46 (63.9%)	208 (70.3%)	
Mean ± SD	11.9 ± 2.6	11.8 ± 2.4	12.0 ± 2.7	0.726
Range	(4–15)	(5–15)	(4–15)	
Site, N (%)				0.369
Head–uncinate process	187 (50.8%)	40 (55.5%)	147 (49.6%)	
Body–tail	181 (49.2%)	32 (44.5%)	149 (50.3%)	
CH-EUS pattern				0.136
Not performed	36 (9.8%)	4 (5.6%)	32 (10.8%)	
Hypovascular	105 (28.5%)	26 (36.1%)	79 (26.7%)	
Isovascular	47 (12.8%)	5 (6.9%)	42 (14.2%)	
Hypervascular	180 (48.9%)	37 (51.4%)	143 (48.3%)	
Needle caliber				<0.0001
25G	251 (68.2%)	69 (95.8%)	182 (61.5%)	
22G	117 (31.8%)	3 (4.2%)	114 (38.5%)	
Needle type				n/a
Standard FNA needle	44 (12.0%)	44 (61.0%)	-	
Side-fenestrated needle	44 (12.0%)	28 (39.0%)	16 (5.4%)	
End-cutting	280 (76.0%)	0	280 (94.6%)	
Number of passes				<0.0001
1–3	307 (83.4%)	28 (38.9%)	279 (94.3%)	
>3	61 (16.6%)	44 (61.1%)	17 (5.7%)	

* Positive value > 39 U/mL; ** size was specified in 349 patients. FNA: fine-needle aspiration; FNB: fine-needle biopsy; SD: standard deviation; EUS: endoscopic ultrasound; CH-EUS: contrast-harmonic endoscopic ultrasound; n/a: not applicable.

**Table 2 diagnostics-14-00427-t002:** Final diagnoses and histological results of endoscopic ultrasound-guided tissue acquisition of 368 small solid pancreatic lesions.

	Overall n = 368	FNA n = 72	FNB n = 296	*p*-Value
Surgery				0.867
Yes	93 (25.3%)	17 (23.6%)	75 (25.3%)	
No	275 (74.7%)	55 (76.4%)	220 (74.3%)	
Final diagnoses				
Benign Malignant	51 (13.9%) 312 (84.8%)	13 (18.1%) 58 (80.6%)	38 (12.8%) 254 (85.8%)	0.249
*Pan-NET*	*206 (66.0%)*	*42 (72.4%)*	*164 (64.6%)*	
*PDAC*	*73 (23.4%)*	*10 (17.2%)*	*63 (24.8%)*	
*Metastases*	*26 (8.3%)*	*2 (3.4%)*	*24 (9.4%)*	
*Solid pseudopapillary tumor*	*4 (1.3%)*	*2 (3.4%)*	*2 (0.8%)*	
*IPMN with dysplasia*	*2 (0.6%)*	*1 (1.7%)*	*1 (0.4%)*	
*Lymphoma*	*1 (0.3%)*	*1 (1.7%)*	*0 (0%)*	
Undefined *	5 (1.3%)	1 (1.4%)	4 (1.4%)	
EUS-TA diagnoses	n = 346 **	n = 62	n = 284	
Benign	52 (15.0%)	12 (16.7%)	37 (12.5%)	
*Normal pancreas*	*23 (44.2%)*	*7 (58.3%)*	*16 (43.2%)*	
*Chronic pancreatitis/Inflammation*	*14 (26.9%)*	*3 (25.0%)*	*11 (29.7%)*	
*Intrapancreatic spleen*	*10 (19.2%)*	*1 (8.3%)*	*9 (24.3%)*	
*Lymph node*	*3 (5.8%)*	*1 (8.3%)*	*2 (5.4%)*	
*Solid serous cystadenoma*	*2 (3.8%)*	*0 (0%)*	*2 (5.4%)*	
Suspicious cells	12 (3.5%)	2 (2.7%)	10 (3.4%)	0.451
Malignant	282 (81.5%)	48 (66.6%)	234 (79.1%)	
*Pan-NET*	*196 (69.5%)*	*40 (83.3%)*	*156 (66.7%)*	
*PDAC*	*56 (19.9%)*	*6 (12.5%)*	*50 (21.4%)*	
*Metastases*	*24 (8.5%)*	*0 (0%)*	*24 (10.3%)*	
*Solid pseudopapillary tumor*	*4 (1.4%)*	*2 (4.2%)*	*2 (0.8%)*	
*Other*	*2 (0.7%)*	*0 (0%)*	*2 (0.8%)*	

* Five patients lost to follow-up after inadequate endoscopic ultrasound-guided tissue acquisition; ** diagnoses on 346 endoscopic ultrasound-guided tissue acquisition samples.

**Table 3 diagnostics-14-00427-t003:** Sensitivity, specificity, positive predictive value, negative predictive value, and diagnostic accuracy of endoscopic ultrasound-guided tissue acquisition in 368 small solid pancreatic lesions using “strict” criteria (i.e., considering “suspicious for malignancy” specimens as negative for malignancy).

	Overall n = 368	EUS-FNA n = 72	EUS-FNB n = 296	*p*-Value
Sensitivity [95% CI]	86.7% [82.5–90.1]	77.2% [65.386.7]	88.5% [83.9–92.2]	0.0125
Specificity [95% CI]	100% [90.7–100]	100% [54.1–100]	100% [88–100]	-
PPV [95% CI]	100% [98.7–100]	100% [93–100]	100% [98.3–100]	-
NPV [95% CI]	46.4% [92.9–99.8]	23.8% [16.9–32.4]	50.8% [42.2–59.4]	<0.0001
Accuracy [95% CI]	88.0% [84.2–91.1]	79.1% [67.9–87.8]	89.8% [85.5–93.1]	0.013

CI: confidence interval; PPV: positive predictive value; NPV: negative predictive value; EUS-FNA: endoscopic ultrasound-guided fine-needle aspiration; EUS-FNB: endoscopic ultrasound-guided fine-needle biopsy.

**Table 4 diagnostics-14-00427-t004:** Sensitivity, specificity, positive predictive value, negative predictive value, and diagnostic accuracy of endoscopic ultrasound-guided tissue acquisition in 368 small solid pancreatic lesions using “non-strict” criteria (i.e., considering “suspicious for malignancy” specimens as positive for malignancy).

	Overall n = 368	EUS-FNA n = 72	EUS-FNB n = 296	*p*-Value
Sensitivity [95% CI]	90.0% [86.2–93]	79.1% [67.4–88.1]	92.8% [88.995.6]	0.0005
Specificity [95% CI]	97.4% [86.1–99.9]	100% [47.8–100]	96.9% [84.2–99.9]	0.130
PPV [95% CI]	99.6% [97.7–99.9]	100% [93.2–100]	99.5% [97.2–99.9]	0.548
NPV [95% CI]	52.8% [44.6–60.8]	26.3% [18.3–36.2]	62.7% [52.1–72.3]	<0.0001
Accuracy [95% CI]	90.7% [87.3–93.5]	80.5% [69.5–88.9]	93.2% [89.7–95.8]	0.0009

CI: confidence interval; PPV: positive predictive value; NPV: negative predictive value; EUS-FNA: endoscopic ultrasound-guided fine-needle aspiration; EUS-FNB: endoscopic ultrasound-guided fine-needle biopsy.

**Table 5 diagnostics-14-00427-t005:** Univariate and multivariate analyses of factors impacting the diagnostic accuracy of endoscopic ultrasound-guided tissue acquisition in 368 small solid pancreatic lesions.

Variables	Univariate Analysis		Multivariate Analysis	
	Accuracy	*p*-Value	*p*-Value	OR (95% CI)
Gender				
Male	87.1%	0.647	-	
Female	89.1%			
Site of the lesion				
Head–uncinate process	84.5%			
Body–tail	91.7%	0.048	0.061	1.88 (0.96–3.73)
Lesion size, mm				
≤10	92.1%	0.107	-	
11–15	86.2%			
Use of CH-EUS				
Yes	87.9%	1.000	-	
No	88.9%			
Needle caliber				
25-gauge	86.0%	0.121	-	
22-gauge	92.3%			
Number of passes				
1–3	89.9%	0.028	0.954	1.14 (0.54–2.86)
>3	78.7%			
FNA	77.9%			
FNB	90.2%	0.014	0.038	2.10 (1.07–4.48)
Final diagnosis				
Benign	80.4%	0.027	0.001	3.56 (1.82–6.94)
Malignant	90.7%			

CH-EUS: contrast-harmonic endoscopic ultrasound; FNA: fine-needle aspiration; FNB: fine-needle biopsy.

## Data Availability

Data are contained within the article.

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
