# Peer review of "EUS-FNA versus EUS-FNB in Pancreatic Solid Lesions ≤ 15 mm"

_diagnostics, 2024, doi:10.3390/diagnostics14040427_

Round 1

Reviewer 1 Report (New Reviewer)

Comments and Suggestions for Authors

Comments to the author

Thank you for giving me the chance to review this manuscript.

This is interesting study, but severely problem, in the following areas.

<Major>

The author includes a 25G EchoTip procore in the FNA group. However, COOK sells the EchoTip procore as an FNB needle, so it should be included in the FNB group. This study needs to be redesigned.

Author Response

Thank you for your answer. Actually, the 25G Procore is a needle for cytology or cell-block. If you look at the company’s brochure

https://cdnnamsseuspwsprod.azureedge.net/data/resources/ESC-D44045-ENF_M3_1535466632942.pdf),

 the acquisition of a histologic core is not expected using the 25G ProCore. It is registered for cytology or cell-block. Moreover, in our study, the material collected with a 25G ProCore was processed as cell-block, not as histologic samples. However, we understand your concern and specified your point as a study limitation.

Reviewer 2 Report (New Reviewer)

Comments and Suggestions for Authors

General:

This is a manuscript comparing the utility of FNA vs FNB in diagnostic sampling of pancreatic lesions that are small (stated <15mm diameter) retrospectively.

Major Issue:

1.     The references mentioned in the Discussion section as well as cancer staging scheme utilize the small lesion cutoff of 20mm. It is unclear how the cut-off of 15mm would affect the malignant population mix, so including lesions up to 20mm would have been more relatable - particularly considering the potential effect of the yield based on the lesion size. 

2.     Along this line, with the overwhelming majority of SPL lesion (particularly of FNA group with small PDAC subset) being PNET in this series (looking at lesions <15mm), the question remains as to whether differences in accuracy of diagnosis of types of “malignancy”, PNET (or PDAC) with FNA or FNB exist. On a brief review of the Table 2, the sensitivities of FNA vs FNB for PNET are 0.95 and 0.95, and for PDAC are 0.6 and 0.79, respectively (without ROSE) – suggesting some differences depending on the proportion of the tumor type. But the actual number of PDAC is relatively small for FNA.

3.     The question of ROSE is present as mentioned – need to know the difference between the FNA w/ ROSE vs FNB. 

Minor Issue:

1.     Spell check

2.     English flow check

Comments on the Quality of English Language

As above.

Author Response

1. RE: We understand your point. However, from a technical point of view, the cut-off of 15mm seems more relevant because it has been demonstrated that very small lesions are more difficult to sample (<10mm in Siddiqui et al., and <15mm in Crinò et al.) with lower diagnostic accuracy and sensitivity in lesions smaller than 15mm or 10mm. Therefore, including lesions up to 20mm could improve diagnostic yield. However, we understand your concern and we have included your point as a study limitation.

2. RE: Thank you for your comment. To respond to your point, we compared the sensitivity of EUS-FNA/B in PDAC and we found no statistically significant difference (60% vs 79%, p=0.228, Fisher exact test). We reported the results in the text.

3.RE: We are sorry, but we cannot answer your question because ROSE was not available in any case (see line 105). We have also included the absence of ROSE as a study limitation.

Minor issue: RE: Done, Thank you!

Reviewer 3 Report (New Reviewer)

Comments and Suggestions for Authors

This study it is conducted well and with an accurate method even in the details, but  is a retrospective study even if compensated by the number of cases and the comparison between FNA and FNB takes place between 2 numerically disproportionate groups: 72 vs 296 respectively. These 2 limitations make the study less original and significant. However, I believe that it can make a small contribution to supporting the FNB method because it documents its feasibility and safety

Author Response

RE: Thank you for your comment. We added your points to the study limitations.

Reviewer 4 Report (New Reviewer)

Comments and Suggestions for Authors

The paper "EUS-FNA versus EUS-FNB in pancreatic solid lesions ≤ 15 mm" is an excellent experience conducted by the authors in the evaluation between EUS-FNA versus EUS-FNB with the result of an obvious advantage of EUS-FNB in terms of quality and quantities of diagnostics. The work is well written with excellent case studies and excellent statistics; the only missing element for the paper to be worthy of publication is the histological evaluation that allows the reader to visually have a comparison between the material obtained with EUS-FNA and EUS-FNB. It is advisable to add a sector dedicated to histopathological evaluation with the consequent iconographic documentation.

Comments on the Quality of English Language

Minimal revsion.

Author Response

RE: Thank you for your comments. We added a figure showing the material obtained with EUS-FNA and EUS-FNB.

Reviewer 5 Report (New Reviewer)

Comments and Suggestions for Authors

1.          The article provides valuable information about the diagnostic performance of EUS-TA in small pancreatic lesions, emphasizing the superiority of FNB over FNA.

2.          This study is well-structured, and the results are presented clearly.

3.          The limitations of this article included the retrospective design, single-center nature and final diagnosis from surgical specimens in a relatively small percentage of cases. However, those limitations were addressed in the discussion and the study's findings still contribute to the evolving understanding of EUS-TA in the context of small SPLs.

4.          Did the 12 cases with TA results of suspicious cell prove to be a malignant disease nature?

Author Response

RE: Thank you for your comments. Yes, the 12 suspicious cases were proved to be malignant at final diagnosis. We specified this point in the text.

Reviewer 6 Report (New Reviewer)

Comments and Suggestions for Authors

While lesion size is said to influence the performance of EUS-TA, the evidence to date has not shown good performance of EUS-TA for small lesions. While the recent introduction of the FNB needle has improved diagnostic performance, there is little evidence showing the usefulness of the FNB needle for small lesions, and this study will provide important evidence because it was conducted on a large number of patients.

However, since this is not an RCT, it must be noted that various biases may exist in needle selection when comparing FNA and FNB needles.

In other words, the FNA-using group may be selecting in difficult lesions such as small lesions/uncus that would be difficult with the FNB needle. Did the diagnostic performance truly decrease with the FNA needle, or are we just looking at differences in the difficulty of the subject? The results of the current analysis are unclear, and we believe that multivariate adjustment of the clinically important parts and improvement of data presentation are needed.

Major

1) The lesion size was FNA:FNB=11.8:13.3 (P=0.598), and the FNA group targeted smaller tumors, although the difference was not significant. At the very least, these backgrounds should be adjusted and presented. For this reason, Table 1 should not only list the median tumor diameter, but also show the difference in distribution between the FNA/FNB groups by sorting them into <5mm, 5-10mm, and 10-15mm, and Table 5 should be adjusted to include tumor diameter in the multivariate analysis.

(2) Since the number of puncture is a large confounding factor, it should be included in the multivariate analysis in Table 5. The OR of the adjusted FNB should be obtained.

3) The 25G needle is used more in the FNA needle group and the 22G needle is used more in the FNB needle group with significant differences. Therefore, in table5, the 25-gauge and 22-gauge portions should also be included in the multivariate analysis. Since the number of events is relatively large, we believe that more explanatory variables can be substituted.

(4) Although the final, sensitivity is 77.2% and 88.5% for EUS-FNA and EUS-FNB, respectively,

 Please provide the breakdown of tumors that were false-negative for EUS-FNA and EUS-FNB, respectively.

Minor

1) Regarding the definition of Adequacy. The part about "sufficient for diagnosis" is very subjective. A more scientific and objective definition is needed.

(2) Was there a selection criterion for puncture needles? It needs to be stated in the Method.

(3) Statistical analysis.

The method of multivariate analysis should be described in detail. Which method was used and how were the variables selected for the multivariate analysis?

4) Abstract

In the body of the abstract, the number of puncture counts is mentioned in the conclusion, even though the data description of the number of puncture counts does not appear in the text of the abstract. If this is the conclusion, the results of the number of puncture should be included in the abstract.

Author Response

1.RE: We understand your point. However, when measured on EUS (that is the measurement of major interest for the present study), the size of the lesions in the FNA and FNB group was the same (11.8 vs 12.0mm). As suggested by the reviewer, we sorted the lesion in Table 1 according to the size (<5mm, 5-10mm, and 10-15mm) and we found no differences between FNA and FNB. We also added tumor size in the univariate analysis, resulting not significant.

2.RE: Thank you for your comment. We added the number of passes as a variable in the multivariable analysis and reported the modified OR for FNB.

3.RE: Thank you for your comment. The needle size variable was not included in the multivariate analysis because the result was not significant in the univariate analysis.

4.RE: Thank you for your comment. We added the rate of false negatives for FNA and FNB groups in the text.

Minor comments: 

1) Regarding the definition of Adequacy. The part about "sufficient for diagnosis" is very subjective. A more scientific and objective definition is needed.

RE: The definition we used is the one recommended in the White Paper of the American Gastroenterology Association (PMID: 29074447)

(2) Was there a selection criterion for puncture needles? It needs to be stated in the Method.

RE: We added the selection criteria for needle type and size

(3) Statistical analysis.

The method of multivariate analysis should be described in detail. Which method was used and how were the variables selected for the multivariate analysis?

RE: The method of multivariate analysis has been described and how the variables were selected was explained.

4) Abstract

In the body of the abstract, the number of puncture counts is mentioned in the conclusion, even though the data description of the number of puncture counts does not appear in the text of the abstract. If this is the conclusion, the results of the number of puncture should be included in the abstract.

RE: Thank you, we added the results of the number of punctures in the abstract.

Round 2

Reviewer 2 Report (New Reviewer)

Comments and Suggestions for Authors

Minor issue: Delete or edit the first sentence in the Discussion section as this paper does not directly analyze for early detection of PDAC (adenocarcinoma) – it is evaluating for “small pancreatic mass lesion” – the series revealed majority of the small lesion <15 mm to be PNET.

Author Response

The sentence has been deleted. Thank you for your suggestion. 

Reviewer 4 Report (New Reviewer)

Comments and Suggestions for Authors

The authors answered the suggestions proposed and now in it's current version the paper is worthy of publication.

Author Response

Thank you!

Reviewer 6 Report (New Reviewer)

Comments and Suggestions for Authors

This manuscript has been well revised according to the reviewers' comments.

Author Response

Thank you!

This manuscript is a resubmission of an earlier submission. The following is a list of the peer review reports and author responses from that submission.

Round 1

Reviewer 1 Report

Comments and Suggestions for Authors

I congratulate the authors for its very interesting and focused paper regarding EUS-TA in small pancreatic lesions.

In addiction to small corrections in English language required, I would be interesting to comment the concordance between of pNET Ki-67 in the FNA/FNB samples and the surgical samples.

Comments on the Quality of English Language

Please check the PDF file attached.

Reviewer 2 Report

Comments and Suggestions for Authors

This retrospective study reported the utility of FNB needle for <15mm pancreatic solid lesion. The utility of FNB needle was already well studied, and the result of this study also follows the previous studies. Therefore, I could not find the novelty of this study.

In addition, the Tables was lack of data (ex. Clinical onset, Surgery, Final diagnosis, EUS-TA diagnoses of FNA and FNB). The calculation of the number is also incorrect (ex. CA19-9: Not available 221 patients; FNA 55 patients + FNB 165 patients 221 patients).

The background of each needle type is also significantly difference (FNA: 25G 96%, 22G, 4%; FNB: 25G 62%, 22G 38%, p<0.01). Because 25G needle is sometimes difficult to obtain the tissue sample due to the needle size, FNA group can significantly reduce the pathological sensitivity.

Considering the unreliable statistical data, and the different backgrounds of each needle, I cannot agree with the publication of this article.